# Single Center Experience Using Monoclonal COVID-19 Antibodies in the Management of Immunocompromised Patients with COVID-19

**DOI:** 10.3390/microorganisms10122490

**Published:** 2022-12-16

**Authors:** David Klank, Bernd Claus, Raoul Bergner, Peter Paschka

**Affiliations:** 1Medizinische Klinik A, Klinikum der Stadt Ludwigshafen gGmbH, 67063 Ludwigshafen, Germany; 2Zentrum für Hämatologische Neoplasien, Onkologisches Zentrum Ludwigshafen, 67063 Ludwigshafen, Germany

**Keywords:** COVID-19, monoclonal antibodies, vaccination, immunosuppressed patients

## Abstract

The medical care of immunocompromised patients with COVID-19 infection causes major hurdles in the management of these patients in clinical practice. However, poor responses to vaccinations in patients with oncological or autoimmune diseases require rapid action and effective care in this fragile patient population. Monoclonal antibodies (mAb) offer an effective therapeutic option with a favorable toxicity profile. We have retrospectively reviewed the first 100 patients treated with mAb in our clinic and assessed the individual vaccine response, side effects of mAb, hospitalization rate and mortality. None of the outpatients treated with mAb had to be hospitalized. In particular, the third SARS-CoV-2 vaccination had a significant effect on the seroconversion (37.5% vs. 77.8% positive patients) in the entire group of patients studied. No side effects of 3°/4° were observed following mAb administration; the mortality in the entire cohort was 7%. Our data and experience show good effectiveness and a favorable tolerability profile of mAb, supporting the feasibility of this therapy in everyday clinical practice. Of note, in immunocompromised patients, both the vaccination status and success need to be recorded in a systematic manner and taken into account in terms of therapeutic intervention using mAb in case of a SARS-CoV-2 infection.

## 1. Introduction

The corona virus disease 2019 (COVID-19) pandemic, which has already been ongoing for more than two years, has challenged the structure of many healthcare systems and has pushed them in developed and developing countries to the limit. The care of immunosuppressed patients in the context of the COVID-19 pandemic has evolved into a particularly large challenge for doctors and patients. In addition to the generally accepted hygiene rules, vaccination is commonly considered to be the most effective method of preventing a severe course of COVID-19. However, it has become more and more clear that immune responses to COVID-19 vaccinations are not sufficient in the vast majority of immunosuppressed patients. Thus, many of the immunosuppressed patients remain at a high risk for a severe course of COVID-19 despite having received a full series of COVID-19 vaccinations, which would generally trigger a sufficient immune response. In contrast, the administration of monoclonal COVID-19 antibodies (mAb) still offers an effective therapeutic option in this fragile patient population, as their mode of action is independent of the individual’s immunity status. In addition, the favorable tolerability profile of the mAb, lack of clinically relevant drug-drug interactions and easy way of delivering these antibodies have facilitated their use in clinical practice. Here, we report on our experience of using mAb in a series of 100 immunosuppressed patients that were treated at our institution.

## 2. Materials and Methods

We have retrospectively analyzed the data of 100 immunosuppressed patients who tested positive for the severe acute respiratory syndrome coronavirus type 2 (SARS-CoV-2) between April 2021 and March 2022 and have subsequently received mAb. In all patients SARS-CoV-2 infection was confirmed by polymerase chain reaction (PCR; BAG Test. Rapid ViroQ SSARS-CoV-2). Gender, age, underlying disease, the type of immunosuppressive therapy, SARS-CoV-2 vaccination status, time from PCR-positivity to the treatment with mAb, side effects following mAb administration, concomitant antiviral therapy, and death within 30 days from mAb application were recorded. Most (*n* = 80) patients received immunosuppressive therapy due to a hemato-oncological or autoimmune disease. Patients with congenital or acquired immunodeficiency diseases, e.g., patients with an HIV infection or patients undergoing immunosuppressive therapy following organ transplantation, were also included. Before mAb administration, patients were assessed for the presence of SPIKE antibodies in serum. The rationale for mAb selection was based on the virus variant, which was determined by using variant-specific PCR (LightMix KIT SARS Spike 417T 681H). In detail, Bamlanivimab was used in the presence of an *Alpha* variant (B.1.1.7), Casirivimab/Imdevimab in *Delta* (B.1.617.2) variants, and Sotrovimab in *Omicron BA1* (B.1.1.529) variants. The use of this algorithm was assumed to achieve the optimal effectiveness in the individual patient. Other virus variants or sublines (especially *Omicron* BA2, BA4 or BA5) were obviously less relevant within the evaluation period. Additional administrations of mAb in case of a prolonged infection after 4 weeks were not included in the current survey.

## 3. Results

The median age in all patients was 62.4 years (range, 18.7–90.9 years); 47% of the patients were male (See Table 1 for patient characteristics). There was no significant difference in age between the three groups (Hemato-oncological 62.5 years, Autoimmune disease 59.0 years, Others 72.6 years; *p* > 0.05 two sided *t*-test). 

At the time of the SARS-CoV-2 infection, the vaccination status was unknown in 10 patients, 12 patients were unvaccinated and six patients had only received one vaccination. Among the 72 fully vaccinated patients, 48 patients had received at least one booster vaccination. The serostatus was negative in 32 patients (IgG and IgA SPIKE antibodies not detectable), 46 patients had a positive serostatus, and in 22 patients, the serostatus was not determined (Table 2). None of the unvaccinated patients had detectable SPIKE antibodies, so no previously unrecognized SARS-CoV-2 infection can be assumed. The median time between PCR-positivity and therapy was 1 day (range, 0–46 days). The mAb administered were bamlanivimab (*n* = 1), casirivimab/imdevimab (*n* = 16) and sotrovimab (*n* = 83). There were no allergic reactions following mAb administration. Three patients complained of self-limiting diarrhea, Common Toxicity Criteria (CTC) 1°, the day after mAb administration. In one case, an episode of vomiting occurred immediately after administration. Only two patients were treated at the same time with antiviral agents (nirmatrelvir/ritonavir), but no relevant side effects were noted in these patients. Both of these patients were seronegative despite previous triple vaccinations; one patient was on treatment with rituximab and prednisone for granulomatosis with polyangiitis, while the other patient had experienced a vaccination failure due to a congenital common variable immunodeficiency. Both of these patients have been treated primarily in the intensive care unit due to impending respiratory failure and have survived. In both cases, intubation and mechanical ventilation could be avoided.

None of the patients treated with mAb in the outpatient setting were hospitalized due to COVID-19. Overall, seven of all the 100 patients died within 30 days after administration of mAb; five of the seven patients were vaccinated at least twice, and in two patients, the vaccination status was unknown. Four of the seven patients who died suffered from hematological diseases. All four patients received therapy with rituximab and chemotherapy for Non-Hodgkin Lymphoma and were undergoing therapy. Two of the patients who died were immunocompromised due to autoimmune diseases. Both received high doses of prednisolone. One patient was treated concomitantly with rituximab. The seventh patient was immunosuppressed with end-stage renal failure and required dialysis; he died due to severe comorbidities. At the time of death, only one of the seven patients tested negative for SARS-CoV-2 by PCR.

In 34 of the 41 (83%) patients with a haemato-oncological disease, the specific antineoplastic therapy was delayed due to COVID-19. In 29 of the 39 (74.3%) patients with an autoimmune disease, the therapy had to be discontinued. However, it cannot be concluded from the data of our study whether the treatment delays or discontinuations had an unfavorable impact on the course and outcome of the haemato-oncological or autoimmune diseases in these patients.

Fifty-three of all the 100 patients were on glucocorticoid therapy with a prednisone equivalent of at least 5 mg per day (≥5 mg/d). Among the patients with ongoing glucocorticoid therapy, 20 of 53 (38.5%) had a negative serostatus, while in the group without glucocorticoid, only 12 of the 47 (25.5%) patients had a negative serostatus (see Figure 1). 

The third vaccination showed a clinically meaningful effect in both the overall cohort and in the subgroup of patients on glucocorticoid therapy with prednisone equivalent ≥5 mg/d. In the patients on prednisone equivalent ≥5 mg/d, only 36% of the patients had a positive serostatus following two vaccinations. Of note, after a third vaccination, the rate of seroconversion increased to 80%. The same was true for the entire group of immunosuppressed patients, where the proportion of seropositive patients increased from 37.5% after two vaccinations to 77.8% following the third vaccination.

CD20-directed therapy was administered in 27 of the 100 patients. Three of the 27 patients had a positive serostatus; two of the three patients were vaccinated before the start of their therapy. Only one of the 25 patients vaccinated during ongoing CD20-directed therapy showed a seroconversion. A subgroup analysis of the serostatus depending on the drugs used for immunosuppressive therapy is presented in Table 2. In addition to the already described strong, adverse impact of CD20 antibodies on the seroconversion, the treatment with methotrexate (MTX) appeared to be an unfavorable factor for responding to COVID-19 vaccinations; seroconversion was documented in only 36% of the patients receiving MTX. Surprisingly, a positive serostatus was detected in 75% of the patients treated with B-cell maturation inhibitors; however, because of the small number (*n* = 4) of patients, this observation obviously requires confirmation in a larger group of patients.

## 4. Discussion

In line with previous reports, our real-world data confirm the limited efficacy of SARS-CoV-2 vaccinations in immunocompromised patients. In particular, treatment with CD20-directed antibodies strongly impairs the humoral responses to vaccinations, but this is also the case in patients on glucocorticoid therapy [1]. However, limitations in the effectiveness of protective vaccinations have also been previously described for MTX and biologicals [2]. Given the observation that a complete series of three SARS-CoV-2 vaccinations seems to provide additional protection with humoral responses in immunocompromised patients, all immunosuppressed patients should be encouraged to receive a full series of at least three SARS-CoV-2 vaccinations. In addition, further vaccinations might be beneficial to maintain sufficient protection against a severe course of COVID-19 [3,4].

A limitation of our study is the sole observation of the humoral immune responses with no data on specific T-cell responses, which are not routinely assessed in clinical practice. Thus, no conclusions from our data can be drawn regarding the impact of the immunosuppression or of the underlying disease on the vaccination effect mediated by specific T-cell responses. However, it is a well-known fact that glucocorticoids cause T-cell apoptosis, as well as a reduction in T-cell function [5]. Thus, adequate T-cell responses cannot be generally assumed in patients undergoing treatment with glucocorticoids. The impact of rituximab on the T-cell response to vaccination is not fully understood. In a study of 59 patients who were treated with rituximab for autoimmune diseases in a rheumatology clinic, the strong influence of rituximab on the humoral response could also be shown. However, a difference in the T-cell response could not be shown in patients treated with rituximab compared to healthy subjects [6]. Patients on rituximab should therefore not be excluded from vaccinations. If appropriate, patients should have their vaccination status updated prior to initiating an immunosuppressive therapy. Of course, this is true not only for SARS-CoV-2 vaccines, but for all necessary vaccinations. Since a sufficient response to vaccines in immunocompromised patients cannot be predicted reliably, the serostatus of immunocompromised patients needs to be recorded in a systematic manner. 

A mortality of 7% in our cohort of immunosuppressed patients with COVID-19 appears to be higher than in the data reported in the general population [7,8]. Since there was no control group in our real-world observation, we used a comparison with historical data. Registry data on 435 COVID-19 patients with hematologic malignancies reported a COVID-19-related mortality of 22.5% [8]. Mortality rates between 7% and 19% were indicated for patients with autoimmune diseases and COVID-19 [9]. However, all these studies only included unvaccinated patients. While a decrease in mortality between the first (03–05/2020) and the second (10–12/2020) COVID-19 wave was noted, to the best of our knowledge, no sufficient study data currently exists for vaccinated patients with hematological malignancies or autoimmune diseases [7].

In accordance with the previous pivotal studies, no CTC grade 3–4° side effects occurred in our patients, and none of the patients treated in outpatient settings had to be hospitalized due to an increasing oxygen requirement [10]. The limited number of patients in our study does not allow us to draw firm conclusions, including whether this was only due to the mAb administration. However, in the mAb approval studies, a reduction in inpatient admissions was observed for the respective mAb preparations [10,11,12]. Our series supports the use of mAb in clinical practice in immunosuppressed patients with a COVID-19 infection, pointing to their easy implementation and lack of severe side effects. 

Effective disease control is one of the most important factors to prevent a severe course of COVID-19, especially in patients with hemato-oncological malignancies or under immunosuppression. Since March 2022, the fixed combination of the mAb tixagevimab + cilgavimab (EVUSHELD) has been approved as pre-exposure prophylaxis to prevent severe COVID-19 infection. In the approval-relevant PROVENT study, a relative risk reduction of 77% for a severe COVID-19 infection was demonstrated for patients using tixagevimab + cilgavimab prophylaxis [13]. The application mode is intramuscular and should be refreshed every 6 months. This mAb preparation obviously offers additional protection in patients who have been on immunosuppressive therapy or rituximab, suggesting it should be used generously in patients in whom the start of the specific therapy cannot be delayed until the completion of a successful vaccination. 

Since the particular effectiveness of mAb has especially been described in seronegative patients, the knowledge of the serostatus of immunosuppressed patients is important for decision-making in terms of mAb administration, particularly in light of the increasing availability of prophylactic mAb depot preparations [11]. Further studies using mAb in immunocompromised patients are required to assess whether their use might lead to a reduction in mortality from COVID-19. Of note, newly arising virus variants pose an additional challenge. While effective mAb were available for virus variants arising during the period of our data collection, to date, no effective mAb is available given the rapid development of Omicron subvariants. Of note, the two latest variants, BQ1.1. and XBB, are also resistant to the latest available mAb (Bebtelovimab) [14]. This also emphasizes the urgent need to perform virus variant-specific PCR to identify the virus subtype, allowing for the adaptation of the mAb therapy.

In our daily clinical practice, we started to use the tixagevimab + cilgavimab prophylaxis in all patients, ideally before they began to receive rituximab therapy, and to refresh the administration after 6 months. In patients who are already undergoing immunosuppressive therapy or therapy with rituximab or other CD20 antibodies, such as e.g., obinutuzumab, the serostatus is determined first. If a patient shows a missing or insufficient titer, a prophylaxis with tixagevimab + cilgavimab is recommended. All patients on some sort of immunosuppressive therapy are advised to contact us in the case of a COVID-19 infection in order to check whether mAb administration is necessary and/or recommendable and to discuss how to proceed, e.g., whether to discontinue the immunosuppressive medication. One obstacle for all hospitals is currently the therapeutic use of mAbs in inpatient care. Since all relevant mAb preparations are now approved drugs, they are no longer provided free of charge by the federal government and need to be bought for inpatients regularly through the hospital pharmacy. However, the health system currently does not provide regular re-imbursement for these quite expensive drugs in in-patient care. Thus, to enable all immunosuppressed patients with a COVID-19 infection to receive mAb without putting economic pressure on the individual hospitals, this situation needs to be resolved quickly. Oral antiviral preparations are a possible alternative here, but many drug-drug interactions and problems with use in case of a renal failure often limit their application. An additional therapeutic option is the use of convalescent plasma (CCP). CCP has been used in other viral diseases with varying results. The effectiveness of CCP in COVID-19 is currently a matter of debate. A study of 160 outpatients showed a reduced progression of the disease when high-titer CCP was administered early during the course of the disease [15]. A study by Tebas et al. showed an improvement in symptoms and 28-day mortality in 80 hospitalized patients with COVID-19 [16]. In contrast, in a meta-analysis including a total of eight studies and 2341 hospitalized patients, no benefit could be shown from the use of CCP [17]. As with mAb, the use of CCP appears to be beneficial, particularly in the early stages of disease in unvaccinated or seronegative patients. Since most CCP originates from an infection with an *alpha* or *delta* virus variant, the efficacy of CCP might be limited due to the currently prevailing multiple *omicron* subtypes. Thus, a determination of the virus subtype needs to be carried out in individual patients before the potential use of CCP. 

The administration of mAb might be beneficial in immunosuppressed patients in terms of preventing a severe course of COVID-19. With the appropriate preparations and training of patients and staff, mAb can be integrated very well into clinical routines in both “inpatient” and “outpatient” settings [18]. Of course, a clinically meaningful impact of mAb in immunosuppressed patients should be assessed in randomized trials, in particular whether mAb administration in this fragile patient population might help to avoid unnecessary and patient-endangering breaks of specific therapies for the underlying disease.

## Figures and Tables

**Figure 1 microorganisms-10-02490-f001:**
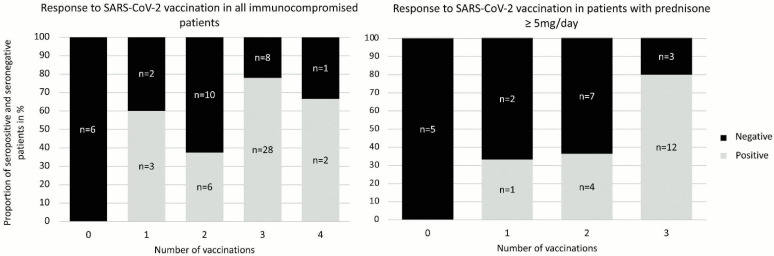
Seroconversion in immunocompromised patients in relation to the number of vaccinations. Only patients with known vaccination and serostatus were included in the figure.

**Table 1 microorganisms-10-02490-t001:** Patient characteristics.

	All Patients		Hemato-Oncological Disease		Autoimmune Disease		Others	
	Outpatient	Inpatient	Outpatient	Inpatient	Outpatient	Inpatient	Outpatient	Inpatient
Sex (m/f)	23/31	24/22	9/9	17/7	7/18	5/8	7/4	2/7
Age (median, years)	62.2	63.3	54.4	73.1	72.0	42.2	55.7	81.8
Vaccination status (unvaccinated/1/2/3 or more vaccinations)	4/3/10/28	8/3/14/17	2/1/7/6	2/2/9/10	2/1/1/16	3/0/3/4	0/1/2/6	3/1/2/3
Prednisone equivalent ≥5 mg/day yes/no	27/27	26/20	9/9	13/11	16/9	11/2	2/9	2/7
CD20 antibody yes/no	13/41	14/32	9/9	11/13	3/22	3/10	1/10	0/9
Time to mAB therapy (days)(Min–Max)	2.6 (0–7)	2.1 (0–46)	1.5 (0–4)	3.7 (0–46)	2.3 (0–7)	2.0 (0–22)	2.4 (0–5)	1.0 (0–3)
Active immunosuppressive treatment yes/no (>6 month)	42/12	31/15	14/4	18/6	23/2	12/1	5/6	1/8
Death	0	7	0	4	0	2	0	1

**Table 2 microorganisms-10-02490-t002:** Serostatus depending on the type of immunosuppressive therapy (SPIKE antibody serum titer: negative, IgG ratio < 0.8; borderline, IgG ratio ≥ 0.8 and < 3.2 (ratio considered positive, but insufficient); positive, IgG ratio ≥ 3.2. Abbreviations: TNFα, tumor necrosis factor; MMF, mycofenolate mofetil; MTX, methotrexate).

	SPIKE Antibody Serum Titer		
Mechanism of Action/Active Substance	Negative *n* (%)	Borderline *n* (%)	Positive *n* (%)
All patients (*n* = 78)	32 (41)	5 (6.4)	41 (52.5)
Prednisone equivalent ≥5 mg/day (*n* = 42)	19 (45.2)	4 (9.5)	19 (45.2)
CD20 antibody (*n* = 16)	12 (75)	1 (6.3)	3 (18.8)
TNFα-inhibitor (*n* = 8)	4 (50)	0	4 (50)
B cell maturation (*n* = 4)	0	1 (25)	3 (75)
MMF (*n* = 5)	3 (60)	0	2 (40)
MTX (*n* = 11)	7 (63.6)	0	4 (36.4)

## Data Availability

The data presented in this study are available on request from the corresponding author. The data are not publicly available due to restrictions (privacy).

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
