# Peer review of "Single Center Experience Using Monoclonal COVID-19 Antibodies in the Management of Immunocompromised Patients with COVID-19"

_microorganisms, 2022, doi:10.3390/microorganisms10122490_

Round 1

Reviewer 1 Report

The manuscript “Single center experience in the management of immunocompromised patients with COVID-19” present data on 100 patients treated with three different monoclonals over 11 months from April 2021 to March 2022. None of the outpatients required subsequent hospitalization and they report seroconversion from 38% to 78% in their population with a 3rd vaccine dose. The paper is valuable for single center characterization of therapy in immunocompromised patients.

The paper will greatly benefit with a table of patient characteristics. In particular, how many were outpatients? Were all the deaths in those treated in the hospital? What was time interval from onset of illness in those treated in outpatient compared to those in the hospital. What is the age sex vaccination status, comorbidities, steroid use, antineoplastic therapy use in the two groups that of inpatients treated and outpatients treated. 7 deaths and no hospitalizations points to high mortality in those hospitalized but the denominators is lacking

Minor

There are published references on additional boosters and seroconversion in this population which should be referenced.

Text page 2 line 69 notes 30 patients with negative serostatus but table 1 notes 31. The table adds up to 77 patient leaving 23 out of 100 but text line 70 page two mentions 29.

Regardless a division between outpatient and inpatients in tables and results would greatly improve this manuscript.

Need to mention omicron Variants BQ.1.1 and XBB and in vitro resistance to existing authorized monoclonals. Drugs and COVID-19 convalescent plasma are indicated in the immunosuppressed.

Author Response

The manuscript “Single center experience in the management of immunocompromised patients with COVID-19” present data on 100 patients treated with three different monoclonals over 11 months from April 2021 to March 2022. None of the outpatients required subsequent hospitalization and they report seroconversion from 38% to 78% in their population with a 3rdvaccine dose. The paper is valuable for single center characterization of therapy in immunocompromised patients.

The paper will greatly benefit with a table of patient characteristics. In particular, how many were outpatients? Were all the deaths in those treated in the hospital? What was time interval from onset of illness in those treated in outpatient compared to those in the hospital. What is the age sex vaccination status, comorbidities, steroid use, antineoplastic therapy use in the two groups that of inpatients treated and outpatients treated.

  • As suggested by the reviewer a table (table 1) showing patient characteristics has been now added into the current version of the manuscript.

7 deaths and no hospitalizations points to high mortality in those hospitalized but the denominators is lacking

  • Appropriate paragraph clarifying the denominators has been included in the current version of the manuscript made (lines 100-109)

Minor

There are published references on additional boosters and seroconversion in this population which should be referenced.

  • We thank the reviewer for this important comment; two references (reference 3 and 4 in the current version of the manuscript) addressing the aspect of additional boosters and seroconversion have been included.

Text page 2 line 69 notes 30 patients with negative serostatus but table 1 notes 31. The table adds up to 77 patient leaving 23 out of 100 but text line 70 page two mentions 29.

  • The discrepancies have been resolved; the original table 1 appears in the current version of the manuscript as table 2.

Regardless a division between outpatient and inpatients in tables and results would greatly improve this manuscript.

  • As suggested by the reviewer the information is now included in the current version of the manuscript the newly created table 1

Need to mention omicron Variants BQ.1.1 and XBB and in vitro resistance to existing authorized monoclonals.

  • A parapgraph addressing the issue of resistance, in particular of BQ.1.1 and XBB virus variants, has now been included in the current version of the manuscript (lines 209-215)

Drugs and COVID-19 convalescent plasma are indicated in the immunosuppressed.

  • As suggested by the reviewer, we discuss the use of covalescent plasma in the current version of the manuscript (lines 234-237)

Reviewer 2 Report

In this study the authors retrospectively reviewed the first 100 patients treated with mAb and assessed the individual vaccine response, side effects of mAb, hospitalization rate and mortality. The data showed good effectiveness and favorable tolerability profile. Some concerns and suggestions are listed as below:

mAb should be mentioned in the title. Please clarify which mAb.

Why the vaccination status was unknown in some patients?

How long these patients had been treated with immunosuppressive agents?

It is not clear for readers if the baseline is different between patients with haemato-oncological diseases and autoimmune diseases.

It is not clear if different therapies have any effects on final results.

Statistical analysis should be improved.

Long-term outcomes are lacking in this study.

Author Response

In this study the authors retrospectively reviewed the first 100 patients treated with mAb and assessed the individual vaccine response, side effects of mAb, hospitalization rate and mortality. The data showed good effectiveness and favorable tolerability profile. Some concerns and suggestions are listed as below:

mAb should be mentioned in the title. Please clarify which mAb.

  • Following the reviewer´s suggestion the title has been modified accordingly.

Why the vaccination status was unknown in some patients?

  • Not all inpatients were cared for in our department. Some patients were only cared for by consultants. Our recommendations were not always implemented. All outpatients were cared for directly by us.

How long these patients had been treated with immunosuppressive agents?

  • A corresponding row was added to the newly created table 1

It is not clear for readers if the baseline is different between patients with haemato-oncological diseases and autoimmune diseases.

  • Information on the baseline characteristics can be found in the newly added Table 1. In addition, a corresponding sentence on age distribution was added (line 69-71)

It is not clear if different therapies have any effects on final results.

  • Given the heterogeneity of our patient cohort and of the therapeutic modalities, no reliable statement can be made in terms the impact of the different therapies on final results given the limited number of patients.

Statistical analysis should be improved.

Due to the sometimes small and heterogeneous patient groups, we only used descriptive methods. We added a comparison of the age distribution between the individual groups. A two-tailed t-test was used here.

Long-term outcomes are lacking in this study.

  • This is a retrospective analysis. At the time of the analysis, a follow-up survey was not intended. However, based on our results, we certainly are interested in long-term outcomes in a potential f/u study.

Round 2

Reviewer 1 Report

The mean or median and the range needs to be noted for symptom onset to infusion.

The statement added on convalescent plasma is not correct. The manufacturing cost is less than monoclonal antibodies. There is both data which shows survival advantage in hospitalized individuals and underpowered studies which do not. Out of all the papers on COVID-19 convalescent plasma the authors chose a small (n=105) study which gave the COVID-19 convalescent plasma late after individual symptom onset of average of seven days with 34% already on mechanical ventilation. Also the current authors failed to mention that the paper demonstrated  in a predefined subgroup analysis which showed a significant benefit of CCP among patients who received a larger amount of neutralizing antibodies. The p value by chi square is 0.05 of high levels of neutralizing antibody with control plasma. The revision should also note a study by Tebas (https://pubmed.ncbi.nlm.nih.gov/34788233/) which did show mortality benefit specifically 28-day mortality (n = 10, 26% vs. n = 2, 5%; P = 0.013) if the current CAPSID trial remains. There are studies which show benefit and studies which do not in hospitalized patients depending on time of administration or dose of antibodies.

Author Response

The mean or median and the range needs to be noted for symptom onset to infusion.

  • Table 1 shows the times from PCR positivity to mAb administration. The time intervals from the onset of symptoms to mAb administration were not recorded and are therefore not available. The range was added to the times from PCR positivity to mAb administration

The statement added on convalescent plasma is not correct. The manufacturing cost is less than monoclonal antibodies. There is both data which shows survival advantage in hospitalized individuals and underpowered studies which do not. Out of all the papers on COVID-19 convalescent plasma the authors chose a small (n=105) study which gave the COVID-19 convalescent plasma late after individual symptom onset of average of seven days with 34% already on mechanical ventilation. Also the current authors failed to mention that the paper demonstrated  in a predefined subgroup analysis which showed a significant benefit of CCP among patients who received a larger amount of neutralizing antibodies. The p value by chi square is 0.05 of high levels of neutralizing antibody with control plasma. The revision should also note a study by Tebas (https://pubmed.ncbi.nlm.nih.gov/34788233/) which did show mortality benefit specifically 28-day mortality (n = 10, 26% vs. n = 2, 5%; P = 0.013) if the current CAPSID trial remains. There are studies which show benefit and studies which do not in hospitalized patients depending on time of administration or dose of antibodies.

  • We thank the reviewer for this comment and clarification. The relevant paragraph has been corrected and adjusted accordingly. The suggested reference has also been added (line 278-288)

Reviewer 2 Report

The authors have addressed my concerns.

Author Response

Thank you for your effort